


# Evaluation of the ERA5 reanalysis as a potential reference dataset for hydrological modeling over North-America

Mostafa Tarek[1], François P. Brissette[1] and Richard Arsenault[1]

[1]École de technologie supérieure, 1100 Notre-Dame West, Montréal, Québec, Canada, H3C 1K3

*Correspondence to*: Mostafa Tarek (Mostafa-tarek-gamaleldin.ibrahim.1@ens.etsmtl.ca)

**Abstract.** The European Center for Medium-Range Weather Forecasts (ECMWF) has recently released its most advanced reanalysis product, the ERA5 dataset. It was designed and generated with methods giving it multiple advantages over the previous release, the ERA-Interim reanalysis product. Notably, it has a finer spatial resolution, is archived at the hourly time step, uses a more advanced assimilation system and includes more sources of data. This paper aims to evaluate the

ERA5 reanalysis as a potential reference dataset for hydrological modelling by considering the ERA5 precipitation and temperatures as proxies for observations in the hydrological modelling process, using two lumped hydrological models over 3138 North-American catchments. This study shows that ERA5-based hydrological modeling performance is equivalent to using observations over most of North-America, with the exception of the Eastern half of the US, where observations lead to consistently better performance. ERA5 temperature and precipitation biases are consistently reduced

compared to ERA-Interim and systematically more accurate for hydrological modelling. Differences between ERA5, ERA-Interim and observation datasets are mostly linked to precipitation, as temperature only marginally influences the hydrological simulation outcomes.

## 1 Introduction

Hydrological science knowledge has long been anchored in the need for observations (Wood, 1998). Observations and

measurements of all components of the hydrological cycle have been used to gain a better understanding of the physics and thermodynamics of water and energy exchange between the land and the atmosphere (e.g. Luo et al., 2018; McCabe et al., 2017; Siegert et al., 2016; Zhang et al., 2016; Stearns and Wendler, 1988). In particular, measurement of precipitation and temperature at the earth's surface has been a critical part of the development of various models describing the vertical and horizontal movements of water. Hydrological models, for example, are routinely used to transform liquid

and solid precipitation into streamflows, using other variables such as temperature, wind speed and relative humidity to increase their predictive skill (Singh and Woolhiser, 2002). Throughout the last several decades, such data has essentially been provided by surface weather stations (Citterio et al., 2015). However, and despite the utmost importance of observed data for hydrological sciences, a net decline in the number of stations in the historical climatology network of monthly temperature datasets has been observed since the beginning of the 21st century (Menne et al., 2018; Lins, 2008). Perhaps

more importantly, data from the NASA-GISS surface temperature analysis shows a particularly large decrease in the number of stations with a long record, a decline starting in 1980. Stations with long records are critical for monitoring trends in hydroclimatic variables (Whitfield et al., 2012; Burn et al., 2012). In addition, the GISS data documents a slow



but consistent decrease in the percent of hemispheric area located within 1200km of a reporting station since the middle of the 20th century (GISS, 2019).

On the upside, other sources of data have steadily appeared to compensate for this worrisome diminishing trend in surface weather stations (e.g. Beck et al., 2018; Sun et al., 2018; Beck et al., 2017a,b; Lespinas, 2016). Interpolated gridded datasets of precipitation and temperature are now common. They allow some information from regions with good network coverage to be extended, to some extent, towards areas with less information. Interpolated datasets, however, do not create new information, no matter how complex and how much additional information is used in the interpolation schemes

(Essou et al., 2016a; Newman et al., 2015). Remotely sensed datasets have long carried the hope of bringing relevant hydrometeorological information over large swaths of land, up to the global scale, and over regions with absent or low-density observational networks (Lettenmaier et al., 2015). There are now several global or near global precipitation datasets derived from various satellites with spatial resolutions varying between 0.125 to 1o (Sun et al., 2018). Ground radar based products are also becoming more common and are available at an even higher resolution (Beck et al., 2019).

All remotely sensed precipitation datasets do however only provide indirect measurements of the target variable. They typically provide biased estimates, and ground stations are often needed to correct the remotely sensed estimates (Fortin et al., 2015).

Atmospheric reanalysis is another product that has generated interest increasingly in the recent decade. Reanalyses combine a wide array of measured and remotely sensed information within a dynamical-physical coupled numerical

model. They use the analysis part of a weather forecasting model, in which data assimilation forces the model toward the closest possible current state of the atmosphere. A reanalysis is a retrospective analysis of past historical data making use of the ever increasing computational resources and more recent versions of numerical models and assimilation schemes. Reanalyses have the advantage of generating a large number of variables not only at the land surface, but also at various vertical atmospheric levels. Data assimilated in a reanalysis consist mostly of atmospheric and ocean data and do not

typically rely on surface data, such as measured by weather stations. Reanalysis outputs are therefore not directly dependent on the density of surface observational networks and have the potential to provide surface variables in areas with little to no surface coverage. Several modelling centers now provide reanalyses with varying spatial and temporal scales (Lindsay et al., 2014; Chaudhuri et al., 2013). Reanalyses and observations share similarities and differ in other aspects (Parker, 2016). Reanalyses have increasingly been used in various environmental and hydrological applications

(e.g. Chen et al., 2018; Ruffault et al., 2017; Emerton et al., 2017; Di Giuseppe et al., 2016). They are commonly used in regional climate modeling, weather forecasting and, more recently, as substitutes for surface precipitation and temperature in various hydrological modeling studies (Chen et al., 2018; Essou et al., 2017; Beck et al., 2017a; Essou et al., 2016b). They have been shown to provide good proxies to observations and even to be superior to interpolated (from surface stations) datasets in regions with sparse network surface coverage (Essou et al., 2017). Precipitation and temperature

outputs from reanalyses have, however, been shown to be inferior to observations in regions with good weather station spatial coverage (Essou et al., 2017). The relatively coarse spatial resolution of reanalyses is thought to be partly responsible for this. Amongst all available reanalyses, many studies have shown ERA-Interim (from the European Centre for Medium-Range Weather Forecasts – ECMWF) to be the best or amongst the best performing reanalysis products (e.g.





Sun et al., 2018; Beck et al., 2017a; Essou et al., 2017; 2016b), arguably the result of its sophisticated assimilation scheme,
and despite a spatial resolution inferior to that of most other modern reanalyses. In March 2019, ECMWF released the fifth generation of its reanalysis (ERA5) over the 1979-2018 period (Hersbach and Dee, 2016). ERA5 incorporates several improvements over ERA-I (see section 3 of this paper).

Of particular interest to the hydrological community are the largely improved spatial (30-km) and temporal (1-hour) resolutions. The spatial resolution is now similar or better than that of most observational networks in the world, with the 75 exception of some parts of Europe and the United-States. The hourly temporal resolution matches that of the best observational networks. In the United-States and Canada, for example, there are currently no readily available observation-derived precipitation and temperature datasets at the sub-daily time scale, and sub-daily records are not consistently available for weather stations. In particular, the hourly temporal resolution, if proven accurate, could open the door to many applications, and notably for modeling small watersheds for which a daily resolution is not adequate. Such 80 watersheds are expected to be especially impacted by projected increases in extreme convective events resulting from a warmer troposphere in a changing climate. Some early results from ERA5 have shown that it outperforms other reanalysis sets and its predecessor ERA-I (Albergel et al., 2018; Olausen, 2018; Urraca et al., 2018).

## 2 Study objectives

This work aims at providing a first evaluation of the ERA5 reanalysis over the 1979-2018 period with an emphasis on 85 hydrological modeling at the daily scale. Even though the hourly temporal scale brings a lot of potential applications for hydrological studies, a first step in the evaluation of ERA5 precipitation and temperature datasets must be performed at the daily scale. The daily scale allows for a comparison against other North-American datasets available at the same temporal resolution, as well as against results from previous studies. In addition, validation at the hourly scale over North-America presents additional difficulties, as discussed above, due to the absence of US or Canadian datasets at this 90 resolution, and to the absence of recorded hourly precipitation for many weather stations. In Canada, for example, fewer than 15% of weather stations have archived hourly variables, and hourly precipitation records contain particularly large ratios of missing data, thus complicating the validation at the regional scale. Consequently, the objectives of this study are to:

1- Provide a first assessment of the potential of ERA5 at providing an accurate representation of precipitation and 95 temperature fields at the daily temporal scale;

2- Evaluate the hydrological modeling potential of ERA5 precipitation and temperature datasets over a large set of hydrologically heterogeneous watersheds using two lumped hydrological models;

3- Based on the above results, document any spatial variability in dataset performance and quantify improvements compared to ERA-I.





## 3 Methods and data

### 3.1 Data and study area

The goal of this study is to evaluate the ERA5 reanalysis product as a substitute for observed data and to compare its properties to those of the older ERA-Interim reanalysis for hydrological modelling uses. Therefore, the ERA5, ERA-Interim and observed (weather station) meteorological datasets were used and basin-averaged over 3138 catchments over Canada and the United-States, whose locations and average elevations are shown in Figure 1. It can be seen that there is a good coverage of the entire domain, although some sparsely populated areas in Northern Canada and in the United-States Midwest have a lower density of hydrometric gauges.

The hydrological models used in this study required minimum and maximum daily temperature as well as daily precipitation amounts. ERA-Interim and the observed datasets were already on a daily time step, however ERA5 is an hourly product and as such, it was necessary to derive daily values from the hourly data by summing precipitations and taking the maximum and minimum one-hour temperatures of the day.

### 3.1.1 ERA-Interim

ERA-Interim (ERA-I) is a global atmospheric reanalysis which was released by the ECMWF in 2006 (Day et al., 2011) in replacement of ERA40. ERA-I introduced an advanced 4-dimensional variational (4D-var) analysis assimilation scheme with a 12-hour time step. It computes 60 vertical levels from the surface up to 0.1 hPa. Its horizontal resolution is approximately 80km. Precipitation and temperature are available at a 12-hour time step and were aggregated to the daily scale in this work. The production of ERA-I will cease in August 2019, thus providing temporal coverage from January 1st 1999 until August 2019.

### 3.1.2 ERA5

ERA5 is the fifth generation reanalysis from ECMWF. It provides several improvements compared to ERA-I, as detailed by Hersbach and Dee (2016). The analysis is produced at a 1-hourly time step using a significantly more advanced 4D-var assimilation scheme. Its horizontal resolution is approximately 30km and it computes atmospheric variables at 139 pressure levels. Data for the 1979-2018 period was released in March 2019. The 1950-1978 period is expected to be released in the summer of 2019. This paper only looks at the 1979-2018 because outputs of reanalysis prior to 1979 have been put into question due to the more limited availability of data to be assimilated, and notably from earth-observing satellites (e.g. Bengtsson, 2004). While ERA5 may solve some of these problems, it is believed that a careful evaluation of inhomogeneity in ERA5 time series would be needed before using pre-1979 data. ERA5 precipitation and temperature was downloaded and aggregated to the daily time step for this work.

### 3.1.3 Observed weather data

The observed weather data come from multiple sources due to the transboundary component in this study. Climate data for catchments in Canada were taken from the CANOPEX database (Arsenault et al. 2016), which includes weather stations from Environment Canada that were post-processed and basin-averaged using Thiessen Polygon weighting. The





data cover the period 1950-2010. Any missing values were replaced by the NRCan interpolated climate data product (Hutchinson et al., 2009).

For the United-States, historical weather data was taken from the Santa-Clara gridded data product (Maurer et al. 2002) as it was shown to be as good as observations for hydrological modelling in a previous study (Essou et al. 2016) and covers a long time period (1949-2010). The data is interpolated along a regular 0.125°x0.125° grid, and is then averaged at the catchment scale.

### 3.1.4 Observed streamflow data

Streamflow records from the United-States Geological Survey (USGS) and Environment Canada were used to calibrate the hydrological models at each of the 3138 catchments and evaluate the hydrological modelling performance. The availability of streamflow data was the limiting factor for the simulation length of many catchments, as it varied from 20 years (minimum amount used in these databases) to over 60 years of streamflow records. Missing data were left as-is and were simply not included in the computation of the evaluation metrics.

### 3.2 Hydrological models

In the course of this study, two lumped hydrological models were implemented and calibrated over each of the available catchments because the large-scale aspect of this study precluded the widespread implementation of distributed models. Although ERA5's spatial resolution is more refined than ERA-Interim (31km vs. 79km), it is still coarse enough that a distributed model would not have changed the results dramatically in this regard. The two hydrological models selected

to evaluate the performance of the various climate datasets, GR4J and HMETS, are flexible, adaptable and have shown to perform well in a wide range of climates and hydrological regimes (Asenault al., 2018; Arsenault et al.,2015, Martel et al., 2017; Valery et al., 2014; Perrin et al., 2003). It was decided to perform the study using two hydrological models in order to assess the impacts of the climate data selection on the overall uncertainty of the hydrological modelling simulations.

### 3.2.1 The GR4J hydrological model

The GR4J hydrological model (Perrin et al. 2003) is a lumped and conceptual model that is based on a cascading-reservoir production and routing scheme. Water is routed from these reservoirs to the outlet in parameterized unit hydrographs. While the original GR4J model includes 4 calibration parameters, the version used in this study had 6 calibration parameters in order to include a snow-accounting and melt routine, namely CEMANEIGE (Valéry et al. 2014). This

GR4J-CEMANEIGE (GR4JCN) combination has shown excellent results in studies across the globe (Huet, M. 2015; Raimonet et al. 2017; Raimonet et al. 2018; Youssef et al. 2018; Riboust et al. 2019; Wang et al. 2019), including in Canada and the United-States. It requires daily precipitation, temperature and potential evapotranspiration (PET) as inputs. The PET was computed using the Oudin formulation (2005) as it was shown to be simple yet efficient when used in GR4JCN. Furthermore, the choice of PET is more sensitive than in other simple hydrological models because GR4J does

not scale the input PET to adjust its overall mass-balance. Instead, a parameter is included that allows exchanges between underground reservoirs of neighboring catchments.



### 3.2.2 The HMETS hydrological model

The HMETS hydrological model (Martel et al. 2017) is more complex than GR4JCN, and as such has more calibration parameters (21). While it is similar conceptually to GR4JCN, it has four reservoirs instead of two (surface runoff,

hypodermic flow from the vadose zone reservoir, delayed runoff from infiltration and groundwater flow from the phreatic zone reservoir) allowing for finer adjustments to the runoff and routing schemes. Its snowmelt module requires 10 of the 21 parameters and was selected specifically to be more robust in Nordic catchments with specific routines for snow accounting, melt, snowpack refreezing, ice formation and soil freezing and thawing. As for PET, it uses the same Oudin formulation as GR4JCN but HMETS includes a scaling parameter on PET to control mass-balance. It has also been used

in large-scale hydrological studies and has shown overall good performance and robustness in a myriad of climates and hydrological conditions.

### 3.3 Hydrological model calibration

As will be detailed in the following section, the three precipitation and three temperature datasets were combined in their 9 possible arrangements for analysis purposes. It follows that the sheer number of calibrations to be performed (3

precipitation datasets x 3 temperature datasets x 2 hydrological models x 3138 catchments) in this study required implementing automatic model parameter calibration methods. For this study, the CMAES algorithm was implemented because of its flexibility (Hansen, et al. 2003). Indeed, it performs well for small and large parameter spaces such as the 6-parameter and 21-parameter spaces in this study. It was also shown to be robust and is considered as one of the best auto-calibration algorithms for hydrological modelling (Arsenault et al. 2014).

The hydrological model parameters were calibrated on the entire available record of data for each catchment, foregoing the usual model validation step. This method was chosen for two reasons. First, calibrating on all years ensures that the maximum amount of information from the climate data is present in the parameter set, and thus that there is no added uncertainty from choosing calibration and validation years. Second, Arsenault et al. (2018) have shown that the model performance is statistically better when more years are added to the dataset, and that validation and calibration skills are

not necessarily correlated.

Finally, the calibration objective function was the Kling-Gupta Efficiency (KGE) metric, which is a modified version of the Nash-Sutcliffe Efficiency metric that was introduced by Gupta, et al., (2009) and Kling et al. (2012). KGE corrects the fact that NSE underestimates variability in the goodness of fit function. It is defined as a combination of three elements:

$$KGE = 1 - \sqrt{(r-1)^2 + (\beta-1)^2 + (\gamma-1)^2} \tag{1}$$

where r is the correlation component represented by Pearson's correlation coefficient, β is the bias component represented by the ratio of estimated and observed means, and γ is the variability component represented by the ratio of the estimated and observed coefficients of variation:

A perfect fit between observed and simulated flows will return a KGE of 1. Using the mean hydrograph as a predictor returns a KGE of 0, and a KGE inferior to 0 implies that the simulated streamflow is a worse predictor of the observed





flows than taking the mean of the observed values. KGE values above 0.6 are generally considered good, however this is a subjective quantification of the quality of the goodness of fit.

### 3.4 Evaluation of the ERA5, ERA-I and observed datasets

The next steps following the calibration of the hydrological models on the 3138 catchments were to analyze the raw climate data (precipitation and temperature) at the catchment scale. This analysis was performed by generating the 9

possible arrangements of 3 precipitation and 3 temperature datasets and comparing their relative differences. Then, after performing the model calibration and hydrological simulation steps, the same type of comparison was performed using the calibration KGE metric as a proxy to the quality of the climate dataset. For example, if a certain combination of precipitation and temperature datasets generates higher KGE calibration scores, it is assumed that the climate data are more likely to be accurate than another dataset that returns lower KGE scores.

The various analyses were conducted on the yearly scale as well as for winter (December, January and February, or DJF) and summer (June, July and August, or JJA) seasons. The results were then analyzed according to their respective catchment locations, climates and sizes in an effort to explain any relationships or differences between the dataset characteristics (i.e. resolution, physics) and their performance (i.e. KGE scores).

## 4 Results

### 4.1 Analysis of precipitation and temperature

The first part of the study was to compare precipitation and temperature values averaged at the catchment scale. Figure 2 shows the mean annual temperatures for the observations, the ERA5 and the ERA-Interim reanalysis products for the catchments in this study (top row). It also shows the mean absolute differences between the datasets for the winter (center row) and summer seasons (bottom row).

The results in figure 2 are averaged at the catchment scale in order to preserve the consistency between the climate data and the hydrological modelling results presented further in this paper. It can be seen that the ERA-Interim and ERA5 temperatures are generally similar to the observations, although ERA-Interim displays a warm bias almost everywhere except for the southeastern United-States and a few catchments in Canada, where it has a cold bias.

On the other hand, ERA5 sees a strong reduction in biases compared to those in the ERA-Interim dataset. The west coast

of North America clearly still shows some important biases of up to 3°C in summer and -2°C in summer, although for most catchments the bias amplitude is smaller. It should be noted that most of the large biases are observed in mountainous areas, where observation networks are generally considered less robust. In the panels representing the differences between ERA5 and ERA-Interim in Figure 2, it can be seen that the ERA5 product corrects the biases in ERA-Interim, i.e. the areas that were too hot in ERA-Interim are colder in ERA5 and vice-versa. The southeast USA was particularly

problematic for ERA-Interim in the context of hydrological modelling (Essou et al., 2016b), and it will therefore be explored further with ERA5 in the rest of this study.





The precipitation time series from the three datasets in this study were compared in a similar manner to the temperature data, with Figure 3 showing the mean annual precipitation for the observations, the ERA5 and the ERA-Interim reanalysis products for the catchments in this study (top row). Figure 3 also shows the mean absolute absolute differences between

the datasets for the winter (center row) and summer seasons (bottom row).

From Figure 3, it is clear that there is a good representation of mean seasonal and annual precipitation values across the study domain. For winter, it seems that ERA-Interim and ERA5 are very similar as the differences between those datasets are small. One exception is the west coast, where a dry bias persists although it has been reduced in ERA5 as compared to ERA-Interim. For the summer period, there is a strong reduction in biases for the eastern half of the United-States

where ERA-Interim was problematic. The dry/wet bias pattern of ERA-Interim is strongly reduced in ERA5. However, both reanalysis products are wet in the North, although as will be discussed in section 5.1, this might be related to the quality of the observation datasets in the remote Northern catchments.

**4.2 Hydrological model simulations**

The first results obtained in the hydrological modelling portion of this study was the performance of the hydrological

models in calibration when driven by the various combinations of precipitation and temperature data. Figure 4 shows the calibration KGE scores for the HMETS (left panel) and GR4JCN (right panel) for the 9 combinations of precipitation (3 sets) and temperature (3 sets). Each boxplot in figure 4 contains the KGE scores of all of the catchments in this study.

From Figure 4, it seems clear that the observations remain the best source of precipitation data for hydrological modelling. ERA5 precipitation is the best reanalysis product, ranking second overall after the observations. It is clear that for

hydrological modelling, the ERA5 dataset is a net improvement over the ERA-Interim reanalysis. For the catchments in this study, using ERA5 precipitation allows reducing the median gap between the older ERA-Interim reanalysis and the observations by approximately 40%. The precipitation data is the main driver behind the differences observed between the datasets as it can also be seen that the variability linked to the temperature dataset is minimal.

Regarding temperature, ERA5 and the observations provide very similar results, whereas ERA-Interim temperature lags

slightly behind. In this sense, the temperature data from ERA5 is marginally more accurate for hydrological modelling at the catchment scale than ERA-Interim, and is similar to that of the observed temperature dataset.

From figure 4, it is also interesting to note that the hydrological models respond similarly to the various inputs, indicating that the improvements seen with ERA5 are due to the dataset rather than the choice of hydrological model. In general, it can also be seen that HMETS performs better than GR4JCN when using the reanalysis datasets (with a median 0.04 KGE

improvement) that is modest but statistically significant using a Kruskal-Wallis nonparametric test. HMETS and GR4JCN are statistically equivalent in terms of KGE when using the observed meteorological data.

The hydrological modelling KGE metrics were next analyzed with respect to the catchment locations, as seen in figures 5 and 6. Figure 5 presents absolute values of KGE efficiency metrics for all three datasets and both hydrological models. The differences between hydrological models (first vs second row) are generally small, although the better performance





of HMETS is particularly clear over the Rocky mountains, and especially in the case of both reanalyses. Both hydrological models perform similarly when using observations as inputs compared to reanalysis.

Focusing on the best performing hydrological model results (first row), two major observations can be made. First, hydrological modeling with observations is clearly superior to using both reanalysis datasets for the eastern part of the US but not so much for Western US and Canada. Second, hydrological modelling performance using ERA5 appears to be
consistently superior to ERA-I. To better emphasize these conclusions, figure 6 presents differences in KGE efficiency metrics between all three datasets. The maps in Figure 6 are therefore obtained by subtracting the maps from figure 5, two at a time. The middle (ERA5) and right (ERA-I) columns present differences in hydrological modeling performance when using reanalyses compared to observations. A blue colour indicates that observations are superior for hydrological modeling, the reverse being true for red colours. This figure provides a clear view of the spatial patterns of hydrological
modeling performance. Observations are clearly superior to renalyses for the eastern half of the US. This corresponds to the zone with relatively large summer precipitation biases presented earlier in figure 3. Outside of this zone, both reanalyses perform similarly to observations, and especially so for ERA5. The left side of Figure 6 testifies to the uniform and significant improvement in hydrological modeling performance when using ERA5 compared to its predecessor ERA-I.

To gain a better understanding of the reasons behind these observations, hydrological modeling performance was analyzed by looking at watershed size (figure 7), elevation (figure 8) and climate zone (figures 9 and 10). In those three cases, the results are only shown for the HMETS hydrological model, since the results for GR4J are similar, albeit with a small degradation in modeling performance, as shown in the preceding figures.

Since all three gridded datasets have different spatial resolutions, figure 7 looks at modeling performance for watersheds
grouped under 4 different size classes. The patterns are consistent across all four size classes, and similar to those of figure 4, with observations being best for all classes, followed by ERA5 and then ERA-I. However, it can be seen that hydrological modelling performance gets progressively better for larger watersheds for all three datasets. This is particularly clear for both reanalyses. While observations perform better at all scales, the gap with reanalysis gets smaller as catchment size increases. The interquartile range (defined by the solid rectangle of the boxplot) is roughly constant for
observations but consistently decreases for both reanalyses. Therefore, a larger proportion of smaller size watersheds are challenging for hydrological modeling than for larger size watersheds. Differences between ERA5 and ERA-I stay constant across all size classes.

Figure 8 presents the same data but as a function of watershed elevation, separated once again in four classes. Mean watershed elevation is mapped in Figure 1. Figure 8 shows a strong dependence of hydrological modeling results on watershed elevation. Observations clearly perform better for the low elevation (< 500 m) watersheds, but differences
rapidly shrink with ERA5 actually performing as strongly and even better than observations for the last two elevation classes. It is relevant to stress that over 60% of all watersheds are included in the first elevation class, and that most of the Eastern US watersheds are within the first two elevation classes. Results from Figure 7 could therefore be influenced by watershed location in addition to elevation. It is also clear that ERA-Interim temperature gets progressively less


competitive as the elevation rises, being significantly less efficient than ERA5 and the observations in the high-elevation groups.

The data was finally analyzed by climate zone groupings. Figure 9 presents North-America's climate classes from the Koppen-Geiger classification (Peel et al., 2007). It can be seen that North-America displays 4 of the 5 main climate zones, with the exception of the Equatorial climate. In total, 13 classes were kept for this analysis. Figure 10 presents hydrological

modeling results for each of those 13 zones.

Results indicate that dataset performance and relative performance strongly depends on the climate zone. This is not surprising since performance was already shown to display spatial patterns. From figures 9 and 10, it is apparent that the ERA5 dataset is systematically better than ERA-Interim for all climate zones and that the observations are clearly superior to ERA5 for the Cfa and Dfa climate zones. Elsewhere, the differences are less pronounced. The Cfa and Dfa climate

zones are the two main climate zones in the eastern US, which were shown to be problematic for the reanalysis datasets. Furthermore, ERA5 fares better than the observations in the Northern parts of Canada and in the mountainous regions with climate zones Dfc and BSh, respectively. This observation will be discussed further, in section 5.2. Figure 11 summarizes these results with the use of the Kruskal-Wallis statistical significance test to determine the best dataset for each climate zone. The Kruskal-Wallis hypothesis test is a non-parametric test to evaluate if two samples originate from

the same distribution. In Figure 11, the green, yellow and red colors respectively indicate the best, second best and worst datasets for each climate zone. If two datasets share a color for the same climate zone, the distribution of KGE values is considered to not be statistically different. Results indicate that there are no differences in hydrological modelling performance between ERA5 and observations over 9 of the 13 climate zones. For the other 4 regions (all in the easternUnited States - Bsk, Cfa, Dfa, Dfb), using observations will result in a statistically significant better hydrological

modelling performance. ERA-I is the worst performing dataset over 8 climate zones. In the remaining 5 zones: Bsh (3), Csa (53), Dsc (33), EF (3) , ET (15)), all three datasets perform identically from a statistical viewpoint. These zones share in common having the fewest watersheds and most extreme climates (arid and polar).

In order to better explore the differences related to the watershed locations and properties, three catchments of different hydrological regimes were analyzed in depth. Figure 12 presents the hydrological modelling KGE difference for HMETS

between ERA5 and the observation dataset (first column) along with the mean monthly precipitation (second column), mean monthly temperature (third column) and mean annual hydrograph (fourth column). Results are presented for the Ouiska Chitto Creek Near Oberlin, Louisiana USA (First row), the Grande Rivière à la Baleine in Quebec, Canada (center row) and the Cosumnes River at Michigan Bar, California, USA (bottom row). Table 1 shows summarized statistics for the three catchments.

The first row in Figure 12 presents a catchment in the southeastern United-States, which is a region in which the reanalysis-driven hydrological models are unable to perform as well as the observation-driven models. ERA-Interim has a clear precipitation seasonality problem, being too dry except for the summer months where there is a large overestimation of precipitation compared to the observations. This seasonality problem is mostly solved by ERA5, but a dry bias persists all year, as shown in Figure 3. The temperatures between the three datasets are practically identical, which means that

evapotranspiration should be relatively constant between the products. The lack of precipitation should therefore become



apparent in the simulated hydrograph, however the streamflow is higher for ERA5 than for the observations when the opposite would normally be expected. It is important to note that the hydrological model can adapt its mass balance by adjusting the potential evapotranspiration scaling, which it has clearly done in this case. The difference in hydrological modelling then comes from the temporal distribution of precipitation, and it can be seen that the ERA5 winter

precipitations are relatively lower in winter than for the rest of the year. The PET scaling therefore attempts to reduce evaporation for the entire year but does not compensate enough to account for this difference in winter. Indeed, it can be seen that the observed hydrograph is underestimated by ERA5 and ERA-Interim for that period in the southeastern United-States.

The second catchment is located in Northern Quebec, Canada, and as such is in a remote and sparsely gauged region. In

this case, it can be seen that the ERA5-driven KGE metric is superior to that obtained using the observations. One key difference between the reanalysis and observed datasets is the precipitation, where ERA5 and ERA-Interim both show more precipitation than the observations. Again, the temperatures are practically identical, meaning that the potential evapotranspiration, although weak in that region, are very similar. The mean annual hydrograph is also very similar between ERA-Interim and the observations, but it can be seen that the ERA5 model overestimates streamflow in winter

while matching the snowmelt peak flows more closely than the other datasets. The difference in KGE in this case comes from a better matching of peak flows, which counts more heavily towards the KGE than the low-flows.

The third catchment, located in the west, is characterized by large precipitation systems in fall and winter, with a months-long dry spell in summer. ERA5 mostly corrected ERA-Interims' strong underestimation of precipitation for that catchment, as is the case for most West-coast catchments as seen in figure 3. ERA5 temperatures are slightly cooler and

are more in-line with the observations. In terms of hydrological modelling, ERA-Interim underestimates the average streamflows year-round while ERA5 slightly overestimates them in winter.  As seen in Table 1, the ERA5 dataset managed to improve the KGE from 0.83 (ERA-Interim) to 0.87, as compared to the reference of 0.90 obtained with the observed data. The improvements in precipitation in ERA5 for this region thus seem to translate to improved hydrological modelling compared to using ERA-Interim, which confirms the findings of figure 6.

**5 Discussion**

This study aims to evaluate the ERA5 reanalysis product as a potential reference dataset for hydrological modelling. The ERA5 reanalysis was compared to the ERA-Interim and observation datasets when used in two hydrological models covering 3138 catchments in North America. This section aims to analyze and explain the results obtained in light of the literature and properties of the ERA5 reanalysis. First, differences in climate and hydrological data will be investigated,

followed by an analysis based on climate classifications and catchment size. Finally, limitations of the study and recommendations for future work will be provided.

**5.1 Differences in temperature and precipitation between the ERA5, ERA-I and observation datasets**

In this study, the observations are taken as the reference dataset and ERA5 is compared to both the observations and ERA-Interim. This allows validating both the improvement in ERA5 with respect to ERA-Interim, as well as evaluating the



possibility of using ERA5 reanalysis data as inputs to hydrological models to overcome potential deficiencies of
observation networks, related to either quality and/or availability.

The evaluation of ERA5 temperature and precipitation variables compared to ERA-Interim and the observation datasets
showed that ERA5 systematically reduced biases present in ERA-Interim for the temperature variables, whereas
precipitation was generally also less biased, although to a lesser degree. There are remaining precipitation biases on the

West coast of North-America with ERA5, but from Figure 2 it can bee seen that the scale of these biases is dependent on
the season. In the Southeast United-States, ERA5 largely corrects biases that were present in ERA-Interim dataset and led
to relatively poor hydrological modelling in a few studies (e.g. Essou et al., 2016b).  As for temperature, Figure 2 shows
that summer temperatures in ERA5 are mostly too high for the catchments west of the Rocky Mountains but are improved
over the ERA-Interim data.

It is important to note that these perceived biases suppose that the observation data is perfect. In reality, at the catchment
scale, one would expect that the observations would be far from perfect and contain errors due to location
representativeness, precipitation undercatch, and missing data due to station malfunction or instrument replacement, for
example. However, the observation data are the best estimates available which makes them the de facto reference dataset.
This means that although Figures 2 and 3 show ERA5 and ERA-Interim as containing some important biases on western

North America, it is possible that these biases are caused by biases in the station data relative to the catchment size. The
reanalysis products also have the advantage of being driven by spatialized sources such as satellites, which can help in
estimating precipitation and temperature data in regions where the weather station network is deficient or sparse.

**5.2 Differences in hydrological simulations using ERA5, ERA-I and observation data as inputs to hydrological
models**

One way to evaluate the quality of the observation and reanalysis data is to use hydrological models as integrators to
compare simulated and observed streamflow, which can act as an independent validation variable. In an attempt to
independently assess precipitation and temperature data for each dataset, all possible combinations of precipitation and
temperature were fed to two hydrological models, which were then calibrated for each combination. This was to remove
any bias caused by parameter sets calibrated on one single dataset, which would obviously be favored in the resulting

analysis. As was the case for the climatological variables, the observed streamflows act as the reference hydrometric data
and are considered as unbiased. Of course, in reality streamflow gauges contain various sources of errors (Baldassarre and
Montanari, 2009), but for this study they are the best available estimates. This hypothesis could have a small effect on the
conclusions of this study. For example, if a certain combination of precipitation and temperature datasets generate higher
KGE calibration scores, it is assumed that the climate data are more likely to be correct than another dataset that returns

lower KGE scores. This could be incorrect in some instances, where the error actually comes from the streamflow data;
however, on average over the 3138 catchments this effect should not influence the results.

The results in Figure 4 showed that the hydrological models driven with the observed precipitation generally provide the
most representative simulated hydrographs, with KGE values exceeding those of the ERA5-precipitation driven
hydrological models by 0.1 on average, which is a significant difference. ERA5 precipitation is also shown to be clearly



better than ERA-Interim precipitation on average for the catchments in this study. Another interesting aspect is that in figure 4, replacing observed temperatures with ERA5 temperatures marginally improves the hydrological modelling skill. While not a significant difference, this attests to the quality of the ERA5 temperatures in general for hydrological modelling. Therefore, the differences observed in the hydrological modelling performance are almost entirely due to the precipitation data quality. The rest of this study will thus focus on the precipitation and hydrological modelling and forego

further analysis on temperature data.

Also of note is that in general, ERA5-driven hydrological simulations are less skillful than those driven by observations. However, there are some catchments - mostly in the mountainous regions of western United-States and in Northern Canada - where use of ERA5 leads to improved hydrological simulations. This is probably due to the difficulty in installing weather stations and obtaining representative observation data in those regions, but it shows that reanalysis data can be

used as a replacement to observations for hydrological modelling in these regions, as previously reported by Essou et al., 2016b).

The more detailed spatial (Fig 6) and climate zone (Figs 10 and 11) analysis outlined the strong spatial dependence on dataset performance. Observations clearly outperformed ERA5 over the Eastern half of the US, where a larger portion of the watersheds used in this study are located. To illustrate this point, Figure 13 presents modelling performance over the

Eastern US (grouping climate zones Cfa, Dfa, and Dfb) against that of the other 10 climate zones.

Figure 13 paints a much different picture than Figure 6 since it shows that hydrological modeling with ERA-5 precipitation and temperature is as good as observations everywhere in North-America, with the exception of the Eastern US. The disproportionate number of watersheds in this region may overemphasize the performance differential between ERA5 and observations as seen in Figure 6. An interesting fact is that the Eastern US is the North-American region having by far the

highest density of weather stations, as reported by Janis et al. (2002). Theoretically, this could explain why observation-based modeling performs better in this region. However, Figure 13 shows that observation-based modelling performance is not different in the other regions, whereas reanalysis-based modeling clearly suffer over the Eastern US. This was also noted in Essou et al. (2016b). It could mean that reanalyses face a harder challenge in the Eastern US, further away from the Pacific Ocean control on atmospheric circulation. A large proportion of summer and fall precipitation in these zones

come from convective storms. Eastern Canadian watersheds are well modelled using reanalyses, but the hydrological behaviour of most of those watersheds is dominated by the spring flood which is largely controlled by temperature, which is very well reproduced by both reanalyses.

Alternatively, this could also mean that Eastern US watersheds are in fact more difficult to hydrologically model and that differences are therefore directly linked to network density. Equal performance of ERA5 and observations elsewhere

would therefore be the result of the improved process representation of ERA5 coupled with some degradation of observations due to the gridded interpolation process between more distant stations. As discussed below, a more precise investigation of modeling performance as a function of station density could shed light on this issue.





### 5.3 Differences between the HMETS and GR4J hydrological models

In this study, two hydrological models were selected to perform the hydrological evaluation of the reanalysis and observation datasets. While both models are conceptually similar, GR4J is simpler than HMETS (two routing processes instead of four, non-scalable PET, much simpler snow model, less than half the number of parameters, etc.). They were shown to perform generally well over all climate zones represented by the catchments used in this study, as can be seen in figure 4. Interestingly, both GR4J and HMETS return similar results for any given driving climate dataset. HMETS performs slightly better than GR4J almost everywhere, although that can be attributed to its more flexible model structure and parameterizations that can better adapt to various hydrological conditions.

Since the main objective of this study was to evaluate the ERA5 dataset for hydrological modelling, the interest is not to compare the hydrological model performances, but to compare the ERA5-driven simulations to the others for each model. In both cases, as can be seen in figure 4, 6 and 8, ERA5-driven hydrological models clearly outperforms the ERA-Interim-driven models, which shows that the precipitation scheme in ERA5 is superior to that in ERA-Interim for hydrological modelling purposes. As stated in section 5.2, temperature seems to play only a minor role in the differences in hydrological modelling.

Furthermore, the observation-driven hydrological models generally perform better than the ERA5-driven models, which confirms that station data should be prioritized when possible. The main caveat to this point is that when the observation station network is of poor quality or too sparse, then ERA5 can be used to fill the voids and get an acceptable hydrological response, as discussed in section 5.2.

### 5.4 Analysis of the impacts of catchment size and elevation on the hydrological simulation performance using the ERA-I and ERA5 reanalyses.

One of the major differences between ERA-Interim and ERA5 is the horizontal resolution, improving from 79km to 31km. This finer resolution should allow for more precise estimations of precipitations and temperatures over smaller catchments that were not adequately represented by ERA-Interim. This logic should apply even though the hydrological models are lumped models. Larger catchments could also see some improvements, namely in a better estimation of the terrain elevation, but it is expected that the gain would not be as large as for smaller catchments.

In order to test this hypothesis, the improvements between ERA5 and ERA-Interim in hydrological modelling were sorted according to catchment size, as shown in figure 7. It is clear from Figure 7 that the catchment size is not a good predictor of hydrological simulation improvement. While most catchments see improvements with ERA5 over ERA-Interim, the catchment size does not seem to affect the rate of improvement. This suggests that the improvements do not come from the higher spatial resolution, lending credence to the hypothesis that the enhancements are due to ERA5's improved physics and process representations.

A similar analysis was performed to evaluate the impact of catchment elevation on hydrological modelling skill. It can be seen from figure 8 that the elevation plays a significant role in the hydrological model's ability to estimate streamflow. For example, the median and interquartile ranges increase for all datasets as elevation increases. This could be caused by





a more rapid hydrological response in higher-elevation and steeper catchments, compared to the slow runoff schemes often found in flat lowlands. The hydrological models being lumped models could contribute to this as large and flat catchments would be more affected by the location of rainfall events compared to steeper ones, especially in the timing

of the hydrograph peaks. For the Northern catchments, the peaks are caused by snowmelt which is much more uniform than rainfall events, which would minimize this effect.

Another, a more probable reason for the reanalysis datasets being stronger in mountainous regions is simply because there are fewer weather stations set up in those areas due to difficulties in accessing and maintaining them. The density of weather stations in the eastern part of the US is typically at least twice as large than for the western part (Janis et al., 2002).

In such cases, a reanalysis would provide information that is not conveyed by station data, making it a de-facto best estimation of precipitation. In essence, the ERA5 data are not yet as accurate as observations, however they are able to perform very well in their absence.

Finally, in all the analyzed scenarios in this study, ERA5 has always been either at least as good as ERA-Interim in terms of hydrological performance. The same is true for the precipitations and temperatures at the catchment scale. From all the

results in this study, there does not seem to be any reason or indication that ERA-Interim should continue to be used for hydrological modelling applications. This is not to say that ERA5 is perfect, but it should become the reference for the time being.

### 5.5 Limitations

As is the case with any large-scale comparison studies, some methodological limitations may potentially impact

conclusions drawn from the presented results. In terms of hydrological modeling, this study only uses two lumped conceptual models and one flow criteria (KGE). Both models are lumped, which limits the assessment of the horizontal resolution component of the three datasets. This aspect was however indirectly assessed by looking at the impact of watershed size. Both hydrological models are conceptually similar but HMETS is more flexible and has more hydrological processes (and parameters). Accordingly, this study was able to look at the impact of parametric space flexibility in dealing

with various datasets biases, but not at other issues such as the impact of physically-based processes and distributed inputs. A study looking at the latter points would require more complex hydrological models, but at the expense of having to look at much fewer watersheds.

The single streamflow criteria and objective function (KGE), like its Nash-Sutcliffe relative, is weighted towards higher flow events. Other objective functions would return different results, however the fact that ERA5 climate data is generally

improved in all areas means that the objective function is unlikely to have a large impact on results. There are several other streamflow criteria which could shed light on differences between datasets, such as extremes. In particular, high flow extremes have the potential to outline improvements in ERA5 compared to its predecessor ERA-I because of improved resolution and processes. Low flows may also be of interest, although the are typically less well-modelled by conceptual hydrological models, and more strongly dependent on temperature, which is very comparable across all three

datasets. Finally, there are now several potential other precipitation datasets that could have been included in the



comparison (see for example Beck et al., 2017a). However, the goal of this work was a first evaluation of the 1979-2019 ERA5 dataset, because of the potential linked to its spatial and temporal resolutions.

### 5.6 Recommendations

One of the main reasons for the interest in the ERA5 reanalysis resides with its hourly temporal resolution. Therefore, the obvious next step is to investigate sub-daily components, and particularly for precipitation. Sub-daily precipitation is key to investigating the hydrological response of smaller watersheds. However, sub-daily studies raise another set of challenges, notably the absence of a robust baseline hourly meteorological dataset.   MSWEP (Beck et al., 2017b) is the best potential candidate at the sub-daily time scale (3-hourly), but the reliability of its sub-daily component is largely unknown. Reliance on hourly weather station data will therefore be required, meaning additional problems including

having to deal with missing data.

The differences noted in Eastern USA raised the question of the potential impact of the density of the station network on the absolute and relative performance of the various datasets.  This could be better studied by assigning a network density index to each watershed. This could ultimately lead to a better understanding of the role of station density, and provide guidance on network improvements or rationalization.

The hydrological performance of ERA5 opens specific avenues of research for streamflow forecasting using ECMWF forecasts. Calibrating hydrological models with ERA5 data could potentially reduce streamflow forecasts biases since the reanalysis and forecasts essentially originate from the same model.

### 6 Conclusion

The main objective of this study was to evaluate the ERA5 reanalysis as a potential reference dataset for hydrological

modeling over North-America, by performing a large-scale hydrological modelling study using ERA5, ERA-Interim and observations as forcing data to two hydrological models. The first assessment showed that ERA5 precipitation and temperature data were greatly improved compared to its predecessor ERA-Interim, although some significant biases remain in the southeast United-States and North-American West coast. These improvements were then shown to translate well to the hydrological modelling results, where both hydrological models showed significant increases in skill with

ERA5 as opposed to ERA-Interim. In all cases, ERA5 was consistently better than ERA-Interim for hydrological modelling, and as good as observations over most of North-America, with the exception of the Eastern half the US. The lesser performance of reanalyses in this region may reflect some deficiencies at representing precipitation seasonality accurately, and may also result from the higher-density network over Eastern USA, thus favoring observations, or a combination thereof. We also showed that the catchment size did not impact the hydrological modelling performance,

thus the improvements are not linked to ERA5's model resolution but to its improved internal physics and assimilation. While some limitations apply to ERA5, it seems that this reanalysis is significantly improved compared to ERA-I and that is should definitely be considered as a high-potential dataset for hydrological modelling in regions where observations are lacking either in number or in quality.


Future work should focus on evaluating the sub-daily performance of hydrological modelling with ERA5, testing its

quality on other continents, integrating ERA5-based model calibration for hydrological forecasting applications and evaluating its potential for weather network augmentation and rationalization.

Finally, it is important to state that this paper does not advocate for the replacement of observed data from weather stations by products such as reanalysis, nor should it be interpreted as providing justification to pursue the current trend of decommissioning additional stations. Weather stations will continue to provide the best estimate of surface weather data

at the local and regional scales and there are many fundamental reasons to keep on supporting a strong network of quality weather stations. The results provided in this study for ERA5 show that atmospheric reanalysis have likely reached the point where they can reliably complement observations from weather stations, and provide reliable proxies in regions with less dense station networks, at least over North America.

## 7 Acknowledgements and data access

This study was partly funded by the National Science and Engineering Research Council of Canada and the Egyptian Armed Forces (Ministry of Defense).

The gridded observed weather data was downloaded from the Santa Clara repository, available here: http://hydro.engr.scu.edu/files/gridded_obs/daily/ncfiles_2010.

The Canopex climate and streamflow data can be downloaded from the official data repository available here:

http://canopex.etsmtl.net/.

The USGS streamflow data can be downloaded from the USGS Water Data for the Nation repository, available here: https://waterdata.usgs.gov/nwis/sw.

ERA-Interim data are available through the ECMWF servers at: https://apps.ecmwf.int/datasets/data/interim-full-daily/.

ERA5 data is available on the Copernicus Climate Change Service (C3S) Climate Data Store: https://cds.climate.copernicus.eu/#!/search?text=ERA5&type=dataset.

The HMETS hydrological model is available on the Matlab File Exchange: https://www.mathworks.com/matlabcentral/fileexchange/48069-hmets-hydrological-model.

Finally, the GR4J model and Cemaneige snow module are made available by the IRSTEA:

https://webgr.irstea.fr/en/models/.

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



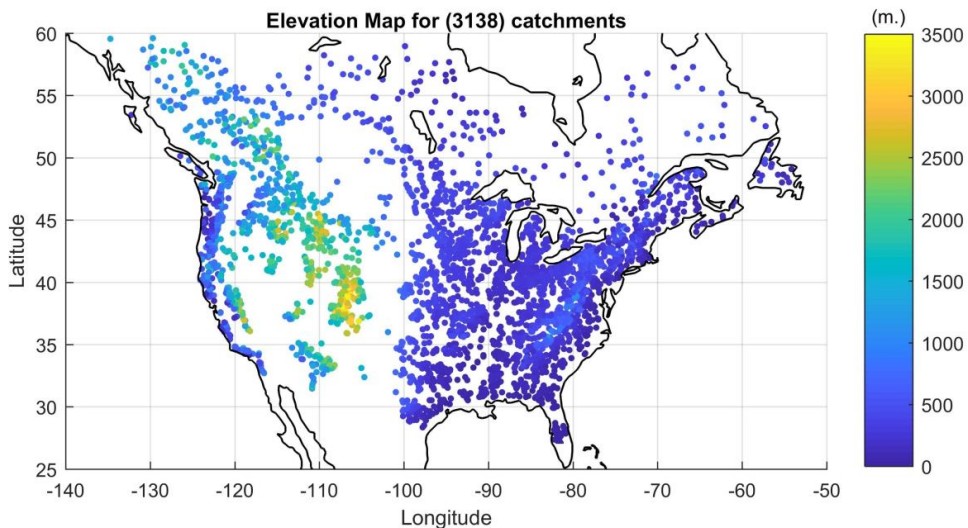

**Figure 1: Watershed locations and their mean elevations over Canada and the United-States.**






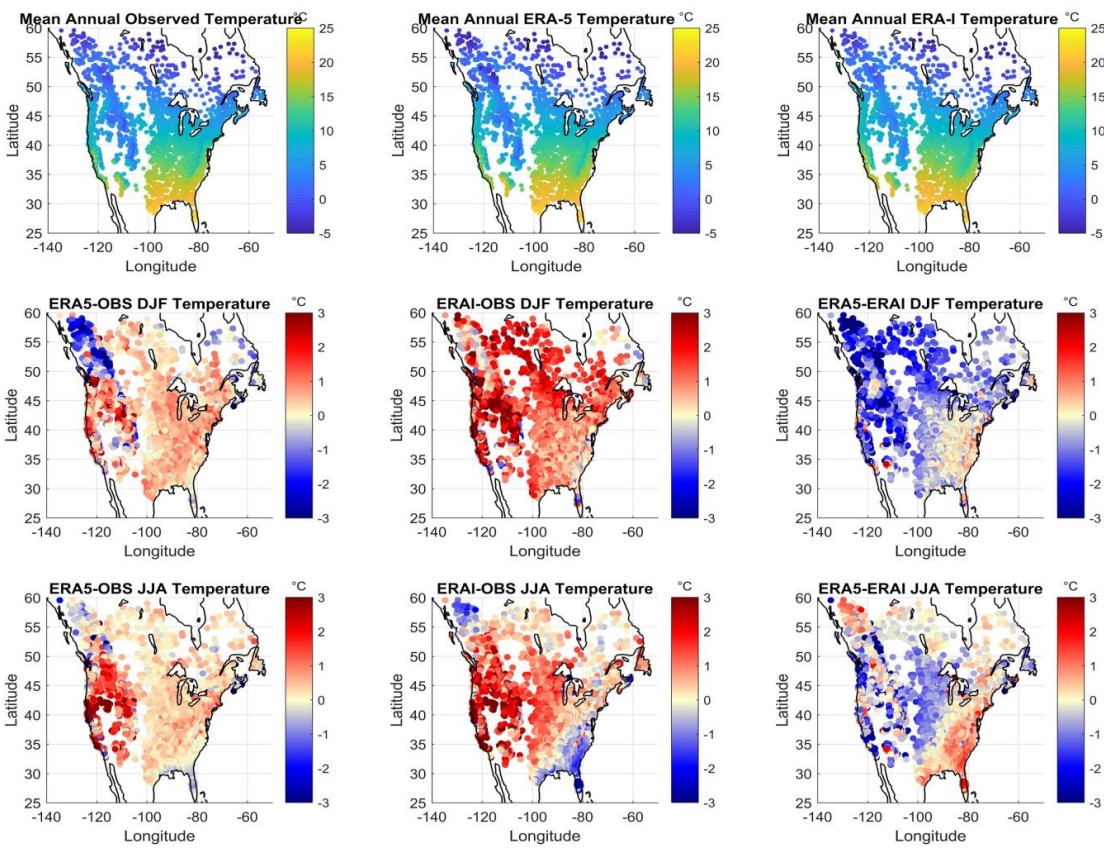

**Figure 2: Mean annual temperature for all three datasets (top row) and seasonal differences (winter in center row, summer in bottom row). All values are in degrees Celsius.**



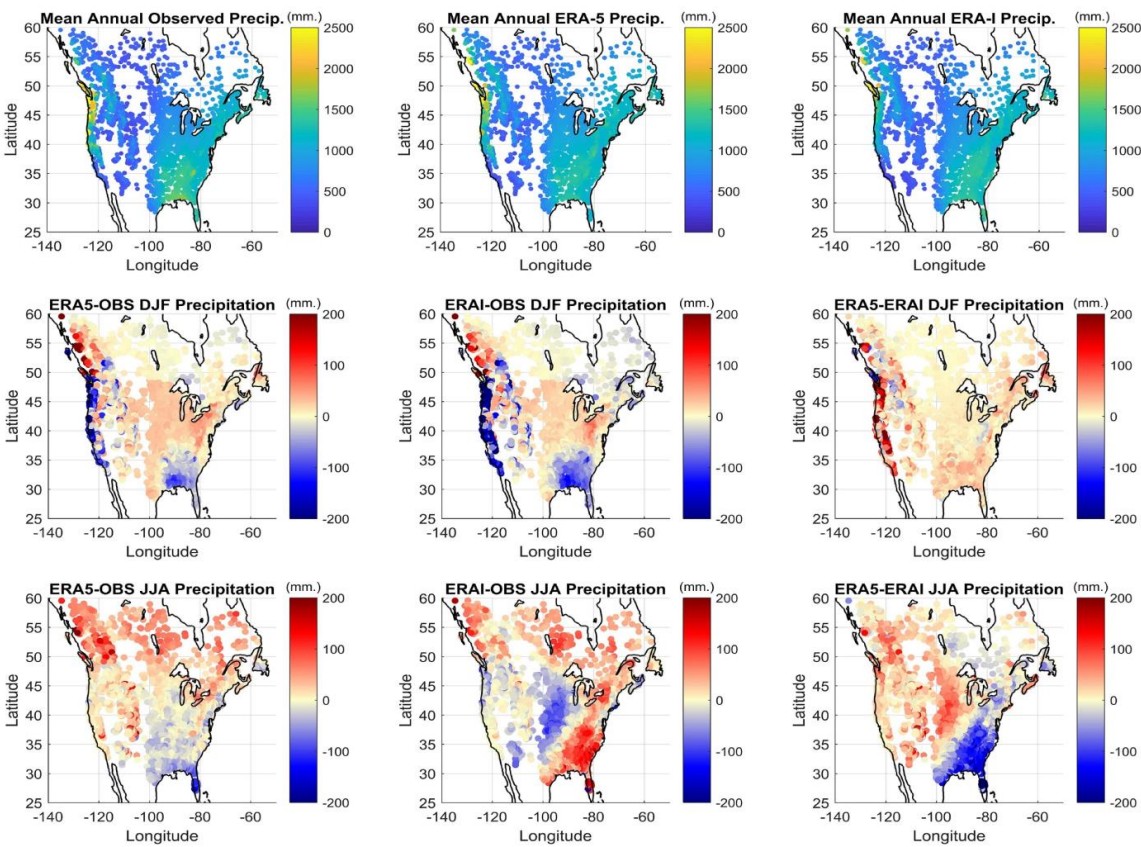

**Figure 3: Mean annual precipitation for all three datasets (top row) and seasonal differences (winter in center row, summer in bottom row). All values are in mm/year.**





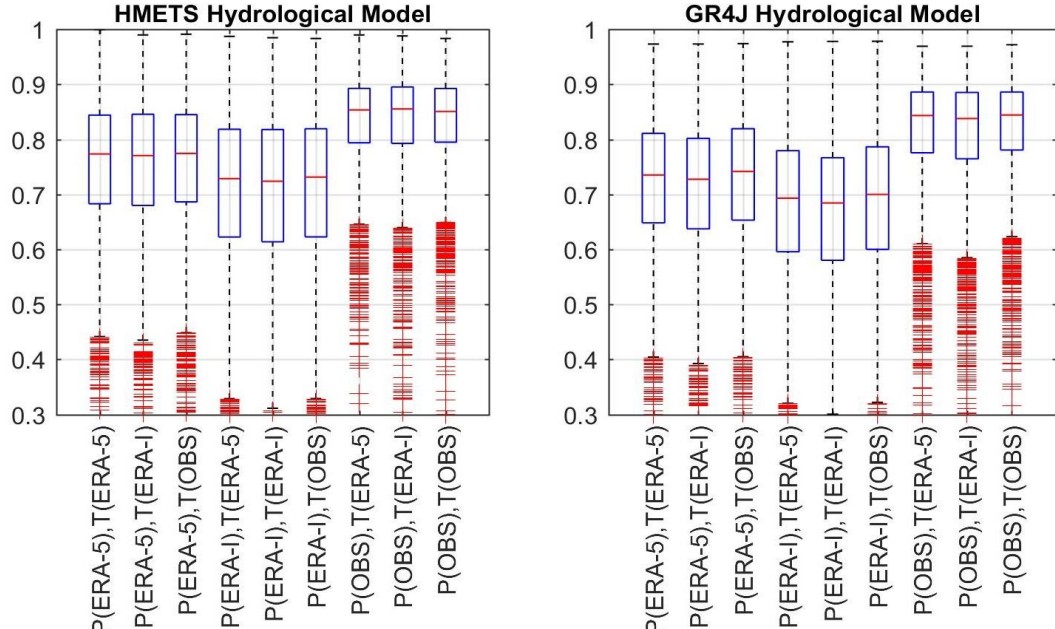

**Figure 4: Distribution of calibration KGE scores for all watersheds as a function of meteorological inputs for HMETS (left panel) and GR4JCN (right panel).**






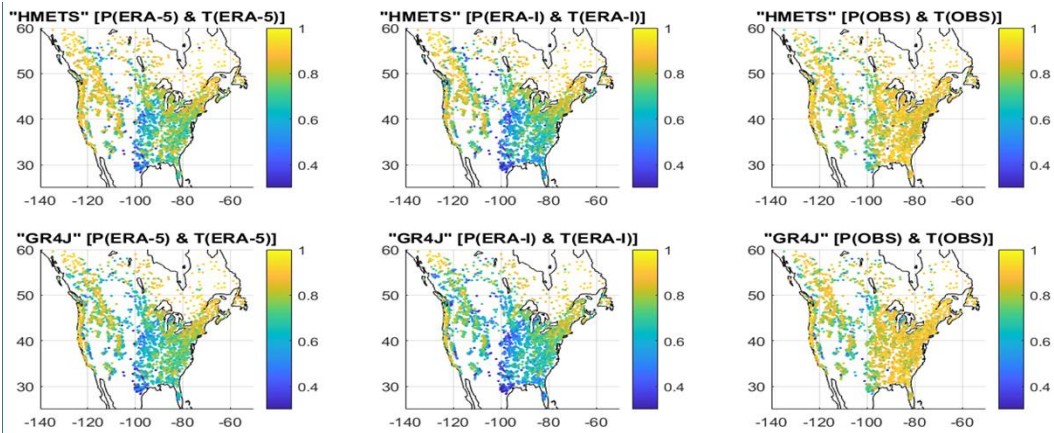

**Figure 5: Spatial distribution of Kling-Gupta efficiency metrics for all 3138 watersheds for the HMETS model (top row) and**
**GR4J model (bottom row), and for ERA5 (left column), ERA-I (center column) and observations (right column).**



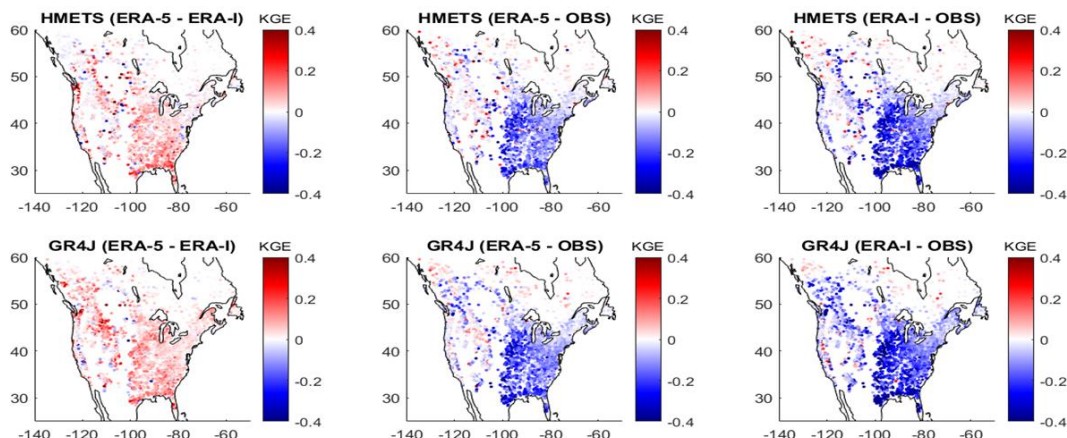

**Figure 6: Spatial distribution of the difference of Kling-Gupta efficiency metrics between the three datasets for all 3138 watersheds, for the HMETS model (top row) and GR4J model (bottom row).**



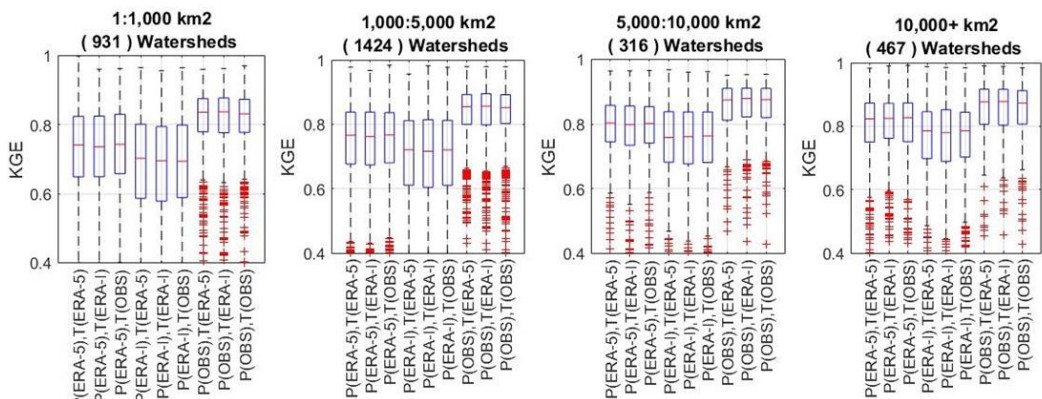

**Figure 7: Distribution of the Kling-Gupta efficiency metrics for various watershed surface areas, for hydrological model HMETS.**



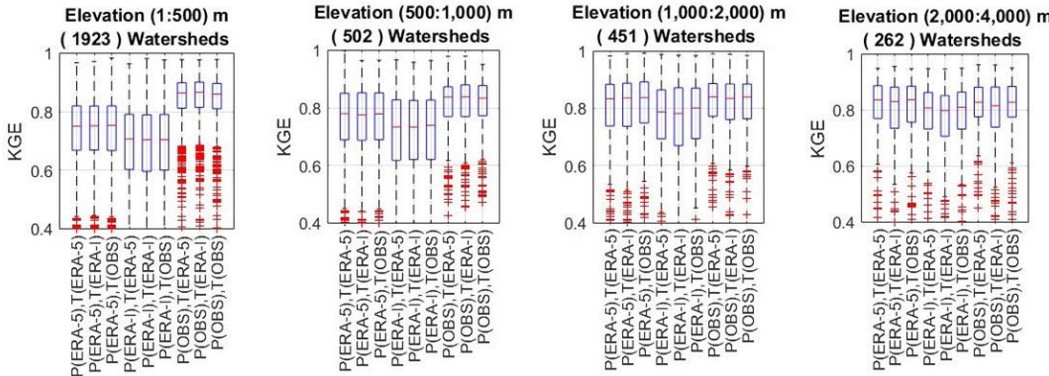

**Figure 8: Distribution of the Kling-Gupta efficiency metrics for various elevation bands, for hydrological model HMETS.**






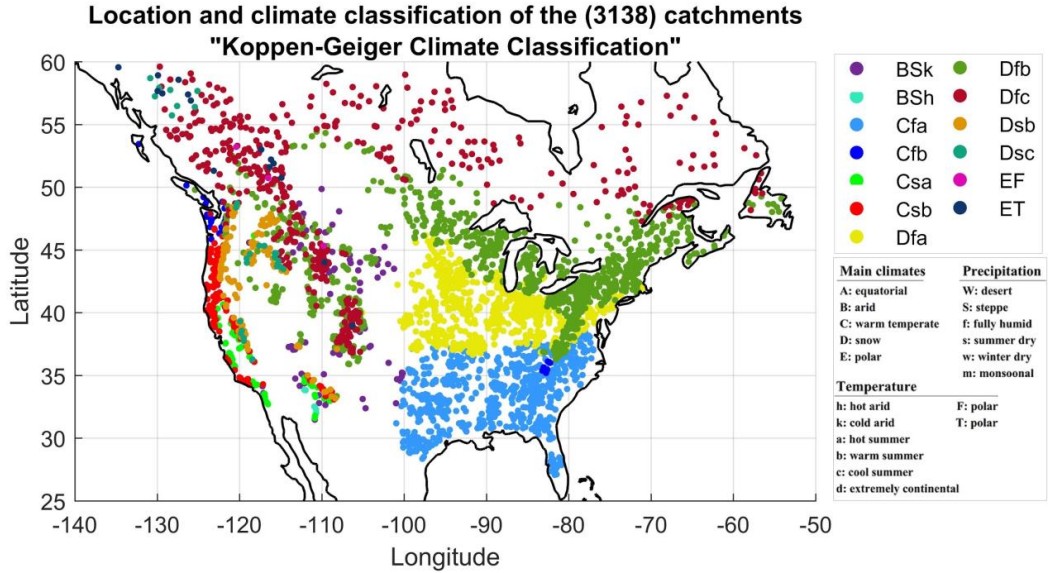

**Figure 9: Koppen-Geiger climate classification of the North-American watersheds presented in this study.**





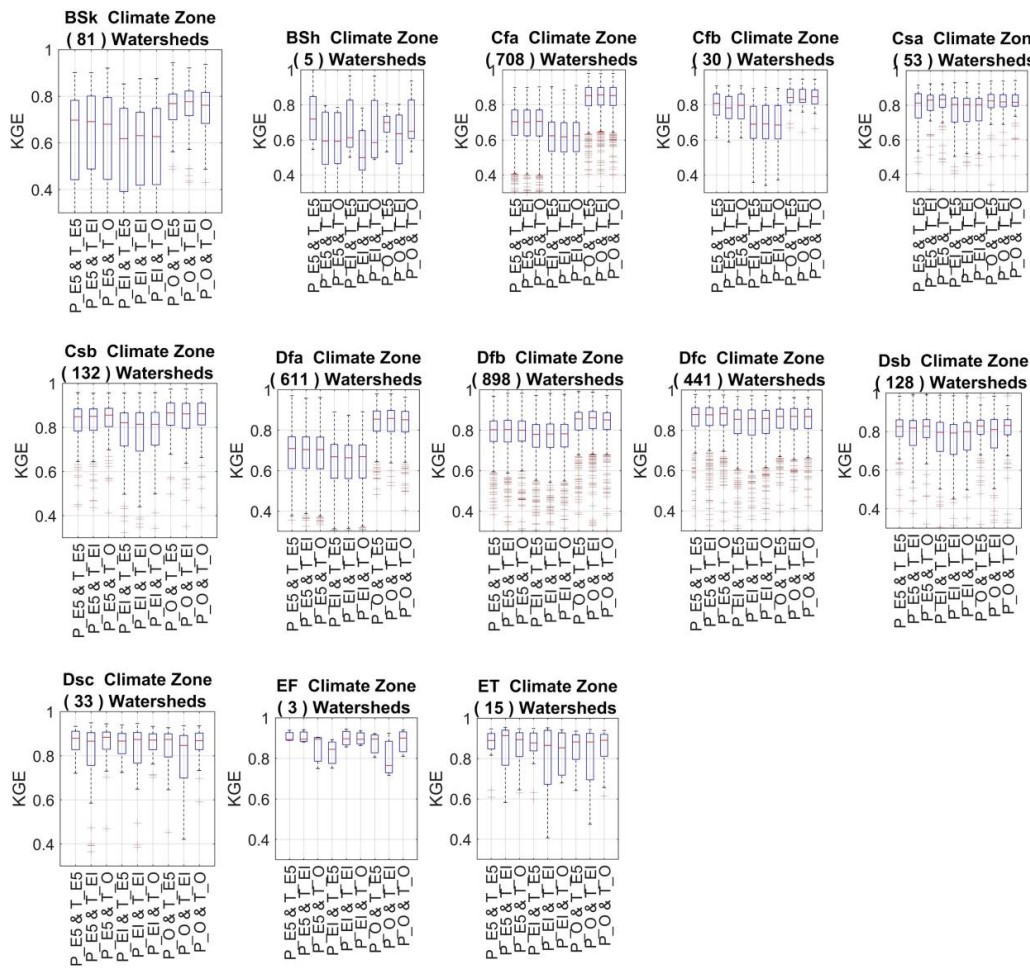

**Figure 10: Distribution of the Kling-Gupta efficiency metrics for the 13 climate zones of Figure 9, for hydrological model HMETS.**



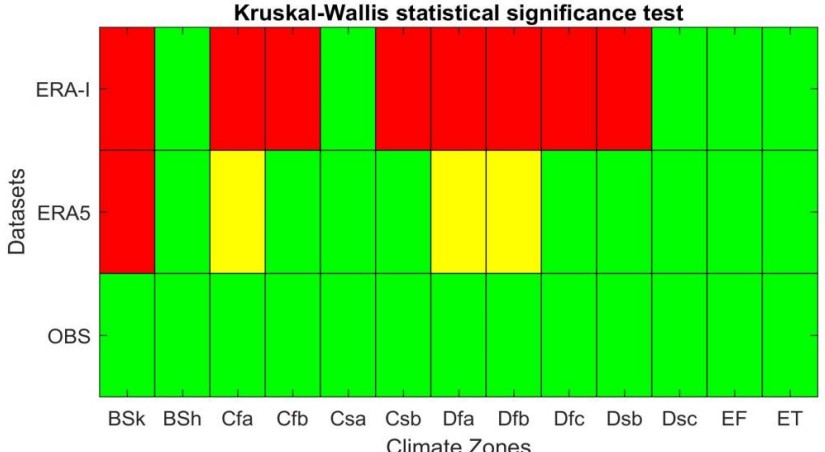

**Figure 11: Results of the Kruskal-Wallis statistical significance test to determine the best dataset for hydrological modelling as observed through the KGE metric, for each climate zone.**


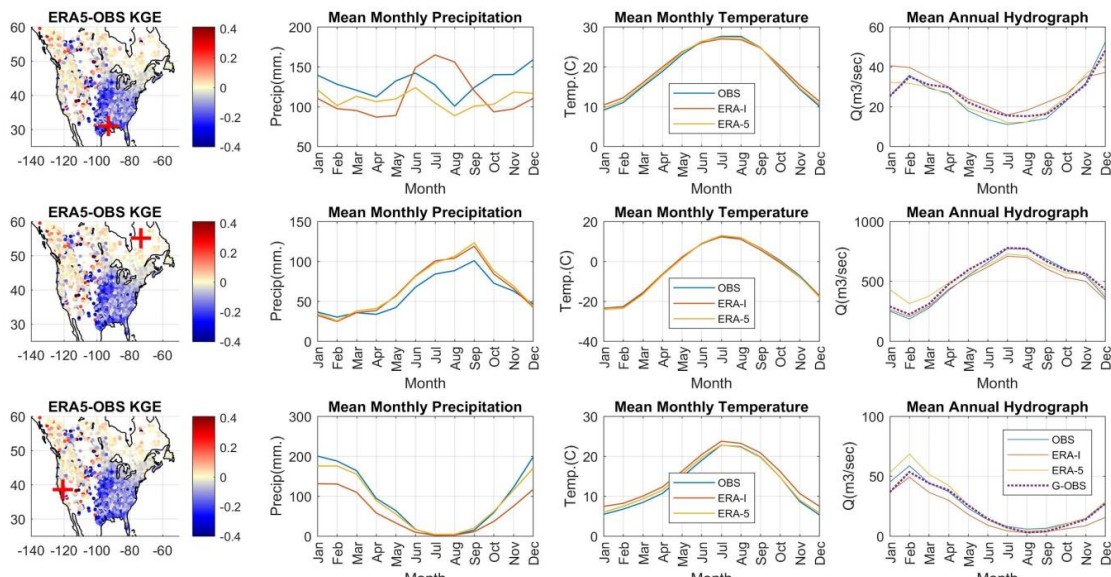

**Figure 12: Difference in hydrological modelling performance, Mean monthly precipitation and temperature and mean annual hydrograph using ERA-I, ERA5 and observations on three dissimilar catchments: Ouiska Chitto Creek (top row), Grande Rivière à la Baleine (center row) and Cosumnes River (bottom row).**




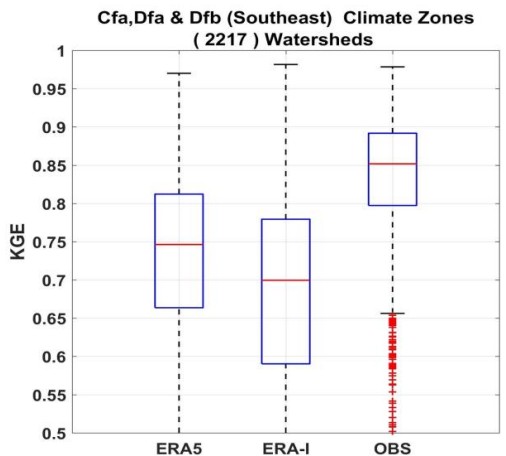
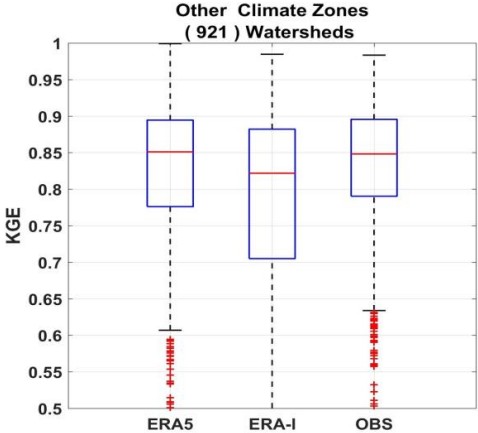

**Figure 13: Distribution of the Kling-Gupta efficiency metrics for the 3 North-East US climate zones (Cfa, Dfa, Dfb) and for all other 10 climate zones grouped together, for hydrological model HMETS.**




**Table 1: Summary of physical and hydrological modelling statistics for the three catchments presented in figure 11.**

| Catchment | Outlet Latitude (dec. deg.) | Outlet Longitude (dec. deg.) | Outlet elevation (m) | Catchment area (km2) | KGE in calibration | | |
| --- | --- | --- | --- | --- | --- | --- | --- |
| | | | | | ERA5 dataset | ERA-I dataset | OBS dataset |
| Ouiska Chitto (Southeast USA) | 30.93 | -92.98 | 53 | 1320 | 0.65 | 0.49 | 0.87 |
| Grande Baleine (Northern Canada) | 55.08 | -73.10 | 389 | 36300 | 0.94 | 0.94 | 0.92 |
| Cosumnes River (Western USA) | 38.60 | -120.68 | 696 | 1388 | 0.87 | 0.83 | 0.90 |