# Peer review of "Evaluation of the ERA5 reanalysis as a potential reference dataset for hydrological modeling over North-America"

_Hydrology and Earth System Sciences, 2019_

## Referee Comment (RC1) · Anonymous Referee #1 · 15 Aug 2019

[referee-annotated manuscript omitted]

---

## Referee Comment (RC2) · Anonymous Referee #2 · 24 Feb 2020

The manuscript evaluates the ERA5 reanalysis as a potential reference dataset for hydrological modelling using two lumped hydrological models in North American catchments. They show, ERA-5 based hydrological modelling performs better and it is equivalent to the observations. Overall, the manuscript is written well and it is well within the scope of HESS. Therefore, I recommend the manuscript for publication, however, with some minor modifications.

I notice some bias pattern between west (cold) and east (warm). Perhaps this may be due to inability of ERA-5 in capturing recent increase in the frequency with which high amplitude ridge trough wave patterns result in simultaneous severe temperature

conditions in both the West and East (singh et al., 2016; Raymond et al., 2017), or with some other reason. It would be good, if the author provides some explanation to this pattern in their revised manuscript.

This study would form a good foundation for those regions where it lacks the observational gauge datasets (such in underdeveloped countries). The authors should add a discussion on this.

References

Raymond, C., Singh, D., & Horton, R. M. (2017). Spatiotemporal patterns and synoptics of extreme wet‐bulb temperature in the contiguous United States. Journal of Geophysical Research: Atmospheres, 122(24), 13-108.

Singh, D., Swain, D. L., Mankin, J. S., Horton, D. E., Thomas, L. N., Rajaratnam, B., & Diffenbaugh, N. S. (2016). Recent amplification of the North American winter temperature dipole. Journal of Geophysical Research: Atmospheres, 121(17), 9911-9928.

---

## Author Comment (AC1) · 11 Mar 2020

**Response to Reviewers' Comments**

We appreciate the efforts of the reviewers and we thank them for their insightful and constructive comments. We have addressed all concerns in this revised manuscript. Below, we provide detailed responses to each of the reviewers' comments. For convenience, we put the reviewer comments in black font, author responses in blue, and direct quotes from the revised manuscript *in italic.*

**Anonymous Referee #1 comments:**

The manuscript provides a relevant evaluation of the ERA5 precipitation and temperature data from a hydrological modelling perspective. The dataset is compared with a previous release of the ERA-type of reanalyses. The manuscript is well-written and uses clear language.
Response:
        We appreciate the reviewer's comments. Please see the point-to-point responses below.

**Specific comments**

[page 3, line 72] ERA5 incorporates several improvements over ERA-I. **Please define acronyms before using them.**
Response:
        We defined the acronyms [page 2, lines 67-68].

        *"Amongst all available reanalyses, many studies have shown ERA-Interim (European Centre for Medium-Range Weather Forecasts (ECMWF) interim reanalysis) to be the best or amongst the best performing reanalysis products (e.g. Sun et al., 2018; Beck et al., 2017a; Essou et al., 2017; 2016b), arguably the result of its sophisticated assimilation scheme, 70 and despite a spatial resolution inferior to that of most other modern reanalyses. In March 2019, ECMWF released the fifth generation of its reanalysis (ERA5) over the 1979-2018 period (Hersbach and Dee, 2016). ERA5 incorporates several improvements over ERA-I (see section 3 of this paper)."*

[page 3, line 86] Even though the hourly temporal scale brings a lot of potential applications for hydrological studies, a first step in the evaluation of ERA5 precipitation and temperature datasets must be performed at the daily scale. **Avoid exaggeration words like "must".**
Response:
        We have modified this in [page 3, lines 86].

        *"Even though the hourly temporal scale brings a lot of potential applications for hydrological studies, a first step in the evaluation of ERA5 precipitation and temperature datasets is performed at the daily scale."*

[page 8, line 249-250] ERA5 precipitation is the best reanalysis product, ranking second overall after the observations. It is clear that for hydrological modelling, the ERA5 dataset is a net improvement over the ERA-Interim reanalysis. **This seems redundant with the previous sentence. Consider merging?**

Response:

Thank you. We have merged the sentences [page 8, lines 250-251].

*"It is clear that for hydrological modelling, the ERA5 dataset is a net improvement over the ERA-Interim reanalysis ranking second after the observations."*

[page 15, line 485] From all the results in this study, there does not seem to be any reason or indication that ERA-Interim should continue to be used for hydrological modelling applications. **..in north America, at least.**

Response:

Thank you. We have modified the sentence [page 15, lines 491].

*"From all the results in this study, there does not seem to be any reason or indication that ERA-Interim should continue to be used for hydrological modelling applications, at least in North America".*

[page 15, line 498] The single streamflow criteria and objective function (KGE), like its Nash-Sutcliffe relative, is weighted towards higher flow events. Other objective functions would return different results, however the fact that ERA5 climate data is generally improved in all areas means that the objective function is unlikely to have a large impact on results. **I believe that KGE is an acceptable objective function, but the justification may need some rephrasing or different argument. I am not convinced that an improvement across regions compensates for an objective function bias toward high flow events. For example if, for every region, a data product is great for high flow events, but terrible otherwise, then an objective function that favours the reproduction of high flow events will deem such a data product to be superior, contrary to an objective function that favours the reproduction of low flow events. The two conclusions would be vastly different. Therefore, observed improvement across multiple regions does not necessarily invalidate a link with the choice of the objective function.**

Response:

We agree that the text was too confident in results that would be obtained with other objective functions. However, we also think that the improvements in ERA5 quality (reduction in biases, etc.) would translate somehow to the quality of simulated streamflow. Therefore, we have rewritten the sentence as follows [page 15, lines 505-507]:

*"Other objective functions would return different results, however the fact that ERA5 climate data is generally improved in all areas is an indicator that other metrics could potentially see improved results as well, although no test has been performed to that effect in this study."*

[page 22, figure 1 title] Watershed locations and their mean elevations over Canada and the United-States.
**Are these points the location of the streamflow gauges rather?**

Response:

Each dot represents the centroid of the watershed and we have clarified that in the figure description.

*"Figure 1: Watershed locations and their mean elevations over Canada and the United-States (each dot represents the watershed centroid)."*

[page 23, figure 2] **Formatting: There is some overlap between the title and the units.**

Response:

Thank you. We have formatted the subplot titles.

[Figure]

[page 32, figure 11] **The legend should be included either in the figure or figure description, instead of having the reader hunt for it in the text.**

Response:

We have edited the figure description to clarify what each color refers to.

*Figure 11: Results of the Kruskal-Wallis statistical significance test to determine the best dataset for hydrological modelling as observed through the KGE metric, for each climate zone. The green, yellow and red colours respectively indicate the best, second best and worst datasets for each climate zone.*

[page 33, figure 12] **What is G-OBS? The streamflow observations? Please define.**

Response:

Exactly, the G-OBS label refers to the streamflow observations. This has been edited in the figure description.

*"Figure 12: Difference in hydrological modelling performance, mean monthly precipitation and temperature and mean annual hydrograph using ERA-I, ERA5, observations (OBS) and streamflow observations (G-OBS) on three dissimilar catchments: Ouiska Chitto Creek (top row), Grande Rivière à la Baleine (center row) and Cosumnes River (bottom row)."*

**General suggestion**

While the superiority of the ERA5 dataset over ERA-interim appears clear, the study lacks independent validation measures and opts to use all available hydrological data for calibration purposes. This choice to use more calibration data is justified versus the traditional reservation of an independent period to validate the calibration. However, while the chance of accidental bias of the calibration method toward a particular dataset is likely slim, it is not guaranteed, or rather the method is not set up to minimize its likelihood. My suggestion to circumvent this issue is to interchange model parameters between ERA5 and ERA-I scenarios. For example, if ERA5 inputs with ERA-I parameters still perform better than ERA-I inputs with ERA-I parameters, this would be a very convincing argument against the impact of the calibration method from being responsible for the perceived improvement of ERA5 over ERA-I.

Response:

Thank you for the suggestion. We have interchanged the model parameters between ERA5 and ERA-I. The results show a significant decrease in the overall performance compared to ERAI_data with ERAI_parameters and/or ERA5_data with ERA5_parameters. This is because the model parameters are strongly dependent on the forcing dataset in the calibration process. Calibration parameters are flexible enough to compensate for the potential biases among the forcing meteorological dataset. Therefore, optimal calibration parameters vary across the different study basins and the selected meteorological datasets (Elsner, Gangopadhyay et al. 2014). However, the model calibrated with ERA5 and driven with ERAI data showed slightly worse results than the model calibrated with ERAI data and driven with ERA5 data. However, we believe these results should not be interpreted in any way to comment on the validity or quality of the reanalysis data.

[Figure]

Reference :

Elsner, M. M., et al. (2014). "How does the choice of distributed meteorological data affect hydrologic model calibration and streamflow simulations?" Journal of Hydrometeorology **15**(4): 1384-1403.

---

## Author Comment (AC2) · 11 Mar 2020

**Response to Reviewers' Comments**

We appreciate the efforts of the reviewers and we thank them for their insightful and constructive comments. We have addressed all concerns in this revised manuscript. Below, we provide detailed responses to each of the reviewers' comments. For convenience, we put the reviewer comments in black font, author responses in blue, and direct quotes from the revised manuscript *in italic.*

**Anonymous Referee #2 comments:**

The manuscript evaluates the ERA5 reanalysis as a potential reference dataset for hydrological modelling using two lumped hydrological models in North American catchments. They show, ERA-5 based hydrological modelling performs better and it is equivalent to the observations. Overall, the manuscript is written well and it is well within the scope of HESS. Therefore, I recommend the manuscript for publication, however, with some minor modifications.
Response:

We appreciate the reviewer's comment. Please see the response below.

I notice some bias pattern between west (cold) and east (warm). Perhaps this may be due to inability of ERA-5 in capturing recent increase in the frequency with which high amplitude ridge trough wave patterns result in simultaneous severe temperature conditions in both the West and East (singh et al., 2016; Raymond et al., 2017), or with some other reason. It would be good, if the author provides some explanation to this pattern in their revised manuscript. This study would form a good foundation for those regions where it lacks the observational gauge datasets (such in underdeveloped countries). The authors should add a discussion on this.

Response:

Thank you very much for these references and the hypothesis. As hydrologists, we are not experts in atmospheric/meteorological phenomena so these guidelines were of help to try and explain these noticeable bias patterns. We have added the following sentences in section 5.1 [page 12, lines 380-384]:

*"There is also an interesting pattern of biases between the East and West coasts (Figures 2 and 3), which could be partly explained by some processes not being accounted for in ERA5, notably the high-amplitude ridge trough wave patterns which have seen a recent increase allowing severe weather in both the East and West simultaneously (Singh et al. 2016, Raymond et al. 2017), although ERA5 did improve the representation of many processes since ERA-I (Hoffmann et al. 2019)."*

We have also added a sentence regarding the use of ERA5 in underdeveloped countries in section 5.6 [page 16, lines 525-527]:

*"It could also be envisioned to extend this work to underdeveloped countries where there is a fewer number of observational gauges, where a good quality reanalysis might allow for improved hydrological simulations and better understanding of the regional weather characteristics."*

**References:**

Raymond, C., Singh, D., & Horton, R. M. (2017). Spatiotemporal patterns and synoptics of extreme wetâAˇRbulb temperature in the contiguous United States. Journal of ˇ Geophysical Research: Atmospheres, 122(24), 13-108.

Singh, D., Swain, D. L., Mankin, J. S., Horton, D. E., Thomas, L. N., Rajaratnam, B., & Diffenbaugh, N. S. (2016). Recent amplification of the North American winter temperature dipole. Journal of Geophysical Research: Atmospheres, 121(17), 9911- 9928.

---

## Author Comment (AC3) · 11 Mar 2020

[revised manuscript text omitted]
  is an indicator that other metrics could potentially see improved results as well, although no test has been performed to that effect in this study. There are several other streamflow criteria which could shed light on differences between datasets, such as extremes. In particular, high flow extremes have the potential to outline improvements in ERA5 compared to its predecessor ERA-I because of improved resolution and processes. Low flows may also be of interest, although the are typically less well-modelled by conceptual hydrological models, and more strongly dependent on temperature, which is very comparable across all three datasets. Finally, there are now several potential other precipitation datasets that could

have been included in the comparison (see for example Beck et al., 2017a). However, the goal of this work was a first evaluation of the 1979-2019 ERA5 dataset, because of the potential linked to its spatial and temporal resolutions.

**5.6 Recommendations**

515 One of the main reasons for the interest in the ERA5 reanalysis resides with its hourly temporal resolution. Therefore, the obvious next step is to investigate sub-daily components, and particularly for precipitation. Sub-daily precipitation is key to investigating the hydrological response of smaller watersheds. However, sub-daily studies raise another set of challenges, notably the absence of a robust baseline hourly meteorological dataset. MSWEP (Beck et al., 2017b) is the best potential candidate at the sub-daily time scale (3-hourly), but the reliability of its sub-daily component is largely 520 unknown. Reliance on hourly weather station data will therefore be required, meaning additional problems including having to deal with missing data.

The differences noted in Eastern USA raised the question of the potential impact of the density of the station network on the absolute and relative performance of the various datasets. This could be better studied by assigning a network density index to each watershed. This could ultimately lead to a better understanding of the role of station density, and provide 525 guidance on network improvements or rationalization. It could also be envisioned to extend this work to underdeveloped countries where there is a fewer number of observational gauges, where a good quality reanalysis might allow for improved hydrological simulations and better understanding of the regional weather characteristics.

[revised manuscript text omitted]